# Outcomes of Treating Tuberculosis Patients with Drug-Resistant Tuberculosis, Human Immunodeficiency Virus, and Nutritional Status: The Combined Impact of Triple Challenges in Rural Eastern Cape

**DOI:** 10.3390/ijerph22030319

**Published:** 2025-02-20

**Authors:** Ntandazo Dlatu, Lindiwe M. Faye, Teke Apalata

**Affiliations:** 1Department of Public Health, Faculty of Health Sciences, Walter Sisulu University, Private Bag X1, Mthatha 5117, South Africa; 2Department of Laboratory Medicine and Pathology, Faculty of Health Sciences and National Health Laboratory Services (NHLS), Walter Sisulu University, Mthatha 5099, South Africa; lfaye@wsu.ac.za (L.M.F.); tapalata@wsu.ac.za (T.A.)

**Keywords:** DR-TB, HIV status, BMI, RR-TB type, TB treatment outcomes, overweight, underweight, nutritional status

## Abstract

Background: Treatment outcomes are critical measures of TB treatment success, especially in resource-limited settings where tuberculosis remains a major public health issue. This study evaluated the treatment outcomes of patients with drug-resistant tuberculosis (DR-TB), co-infected with human immunodeficiency virus (HIV), and the impact of nutritional status, as measured by body mass index (BMI), on these outcomes in rural areas of the Olivier Reginald Tambo District Municipality, Eastern Cape, South Africa. Methods: A retrospective review of 360 patient files from four TB clinics and one referral hospital was conducted between January 2018 and December 2020. Data collected included patient demographics, clinical characteristics, BMI (categorized as underweight, normal, overweight, or obese), HIV status, DR-TB type, and treatment outcomes. Statistical analyses assessed the association between BMI categories, HIV status, and treatment outcomes. A scatter plot was used to illustrate BMI trends as a continuous variable in relation to age, enabling an analysis of BMI distribution across different age groups. Additionally, bar charts were utilized to explore categorical relationships and patterns in BMI across these groups. Results: The majority of patients were co-infected with HIV and had DR-TB, with rifampicin-resistant TB (RR-TB) and multidrug-resistant TB (MDR-TB) being the most prevalent forms. Treatment outcomes varied significantly by BMI category. Underweight patients had the lowest cure rates (23.2%), highlighting the adverse impact of malnutrition on DR-TB treatment success. Patients with normal BMI demonstrated higher cure rates (34.7%), while overweight and obese patients had moderate outcomes. HIV co-infection further reduced cure rates, with co-infected individuals showing poorer outcomes than HIV-negative patients. Gender disparities were also observed, with females achieving higher cure rates (39.1%) compared to males (31.4%). Weak trends linked BMI and DR-TB type, such as a higher prevalence of normal BMI among RR-TB cases. Conclusion: This study underscores the significant influence of nutritional status on DR-TB treatment outcomes, particularly among patients co-infected with HIV. Underweight patients face the greatest risk of poor outcomes, emphasizing the need for nutritional support as a critical component of DR-TB management. Comprehensive HIV care and gender-specific interventions are also essential to address disparities in treatment success. Tailored strategies focusing on these aspects can significantly enhance outcomes in high-burden, resource-limited settings.

## 1. Introduction

In 2022, the World Health Organization (WHO) proclaimed an encouraging message: drug-susceptible tuberculosis is not just a challenge, but a disease that can be definitively cured! This breakthrough offers hope and highlights the power of modern medicine in combating health threats [1]. One of the important global indicators for monitoring the implementation of WHO’s End TB Strategy from 2015 is the treatment success rate, which encompasses both cure and treatment completion according to WHO’s report [2]. In low-income nations, malnutrition and infectious diseases are significant health challenges. The interconnectedness of tuberculosis epidemiology with social and economic conditions has made its prevention and control a challenging goal to achieve [3]. Their coexistence is partly due to poverty being an important determinant of both problems, but also due to the two-way causal interactions between nutritional deficiencies and infections, whereby infections exacerbate nutritional deficiencies, which in turn increase infectious disease morbidity and mortality. Most research has been conducted on the relationship between generalized malnutrition or micronutrient deficiencies and childhood infections [4]. South Africa is in the grip of three concurrent epidemics: malnutrition, brought about by a conglomeration of socioeconomic factors; HIV/AIDS, caused by the human immunodeficiency virus; and active TB, caused by progressive infection *with Mycobacterium tuberculosis*. Although caused by separate factors, there is evidence that each epidemic acts synergistically to aggravate the other two [5]. Tuberculosis is strongly influenced by nutritional status [6], with nutrition interventions being likely to affect the prevalence of active disease, response to drug therapy, and quality of life [7]. An understanding of the basic science of nutrition, immunity, and infectious disease pathogenesis is necessary for the rational identification of appropriate therapeutic interventions [8]. Recent studies investigating the pathogenesis of HIV infection have provided the exciting novel conceptual understanding that the gastrointestinal tract is a major anatomical front line of the disease, and that lymphocyte activation is a key step in the CD4+ T cell depletion that defines AIDS. Together, these insights have major implications for our dawning understanding of the intersection between nutrition and HIV/AIDS, both in terms of the potential impact of HIV infection on nutritional status and in redefining our conceptions of how nutritional intervention might affect HIV/AIDS pathogenesis [5]. Adequate nutrition plays a critically important role in supporting the health and quality of life of people with TB and human immunodeficiency virus (HIV) [6,9]. In our study, the triple burden refers to the concurrent presence of drug-resistant tuberculosis (DR-TB), human immunodeficiency virus (HIV), and obesity, where the combined effects of these conditions may synergistically worsen the patient’s health. Each of these conditions in our study setting presents unique difficulties in terms of their co-occurrence and significantly challenging treatment outcomes. The interplay between TB and HIV creates a vicious cycle where each disease exacerbates the other, leading to a faster progression of HIV and an increased mortality risk, especially in people who are obese. The global proportion of people living with HIV who are overweight (BMI: 25.0–29.9 kg/m^2^) or obese (BMI: ≥30 kg/m^2^) has risen significantly [10]. While obesity does not directly increase the risk of contracting TB or HIV, it can complicate the course of these diseases by impairing the immune response crucial for effective TB and HIV treatment. Drug-resistant TB (DR-TB) arises when the *Mycobacterium tuberculosis* bacterium becomes resistant to commonly used treatments, often due to improper use of TB drugs, incomplete treatment courses, or natural genetic mutations [11]. The presence of DR-TB complicates TB control efforts and heightens the risk of transmission within communities, as observed in our study setting. Drug-resistant tuberculosis refers to TB that is resistant to isoniazid (isoniazid mono-resistance), rifampicin (rifampicin mono-resistance), or both, with the latter known as multidrug-resistant TB (MDR-TB) [12]. When fluoroquinolone resistance is added to MDR-TB, it is classified as pre-extensively drug-resistant TB (pre-XDR-TB), and with additional resistance to bedaquiline and linezolid, it becomes extensively drug-resistant TB (XDR-TB) [12]. Drug-resistant tuberculosis (DR-TB) includes multidrug-resistant tuberculosis (MDR-TB), which is resistant to at least isoniazid and rifampicin, the two most effective first-line anti-TB drugs. It also includes extensively drug-resistant tuberculosis (XDR-TB), which is MDR-TB that has additional resistance to any fluoroquinolone and at least one of the three injectable second-line drugs: amikacin, kanamycin, or capreomycin [12]. DR-TB caused by MDR strains of *Mycobacterium tuberculosis* and extensively DR (XDR) TB strains are emergent problems. People with HIV face an increased risk of contracting TB, making co-infection a significant health concern, especially in areas where both diseases are widespread [13]. TB is a leading cause of death among people living with HIV, and co-infection significantly worsens the course of HIV, accelerating its progression to AIDS [14]. The TB-HIV co-infection is a dual epidemic as each speeds the other’s progress. HIV makes TB more difficult to diagnose and treat because the sputum of an HIV-positive person usually contains a lower concentration of TB bacteria, making it harder to detect in a sputum test [15]. HIV and ART drugs affect the partitioning and distribution of the adipose tissue compartment, which can lead to metabolic problems [10]. TB and HIV co-infection has a synergistic effect, with each disease accelerating the progression of the other and increasing the risk of mortality. Additionally, obesity exacerbates this situation by impairing immune responses through chronic low-grade inflammation, which further complicates effective treatment outcomes [16]. This may impair the body’s ability to fight tuberculosis, resulting in more severe disease progression and consequences in HIV-positive patients [16]. Obesity can cause problems during tuberculosis treatment, such as the need for medication adjustments and increased monitoring for side effects, and this complexity can impede efficient treatment and increase healing timeframes [10]. Obesity increases the burden of metabolic disorders among HIV patients [17]. Obesity exacerbates the treatment outcomes of TB and HIV through multiple interconnected mechanisms. First, obesity induces chronic low-grade inflammation, which impairs immune responses that are critical for controlling TB infection and responding to antiretroviral therapy (ART) for HIV [17]. This state of immune dysregulation may compromise the effectiveness of treatments, leading to slower recovery and higher complication rates. Second, metabolic changes associated with obesity, such as insulin resistance and dyslipidemia, can interfere with the pharmacokinetics of TB and HIV medications, necessitating careful adjustments to treatment regimens. Lastly, obesity often complicates the clinical course of these diseases by increasing the risk of comorbidities, such as diabetes and cardiovascular conditions, which can further hinder treatment adherence and success [16,17]. Obesity is associated with mental health challenges that can impair adherence to TB and HIV treatment regimens and alter the immune response needed for TB containment, while TB itself can disrupt neurotransmitter hormonal balance, further affecting mental health [18]. Poor mental health can lead to decreased motivation for maintaining a healthy lifestyle, increasing the risk of multi-morbidity in overweight and obese HIV patients [19,20]. The global triple burden pandemic of DR-TB, HIV, and obesity is a rising challenge in low- and middle-income countries [10]. The co-occurrence of these diseases represents colliding pandemics, but analyzing TB treatment outcome, patient category, social history, type of regimen, age and gender influences, and clinical characteristics at the start of treatment for these three major diseases in rural areas of Eastern Cape has not been reported in the literature. The coexistence of these three conditions creates a complex web of challenges for healthcare providers as well. DR-TB and HIV require specialized treatments and monitoring, while obesity necessitates a focus on lifestyle and metabolic health. In rural areas of Eastern Cape, TB-HIV co-infection is very common [21]. HIV and obesity significantly influence DR-TB treatment outcomes, necessitating the close coordination of treatment strategies and addressing the complex relationship between HIV and TB management [22]. The effect of comorbidities induced by both HIV and obesity on DR-TB treatment outcomes is not well established in rural areas of the Eastern Cape. Hence, this study was designed to assess the treatment outcome of patients with DR-TB comorbid with HIV and obesity in comparison with non-HIV and non-obese infected individuals.

## 2. Materials and Methods

### 2.1. Study Design

This study involved a retrospective evaluation of clinical data from patient files collected between January 2018 and December 2020. It was conducted at four TB clinics and one referral hospital located in the Oliver Reginald Tambo District Municipality, Eastern Cape, South Africa. These clinics were chosen because they serve as primary care facilities for diagnosing and managing tuberculosis (TB), particularly drug-resistant TB (DR-TB), in rural and underserved communities that face a high burden of disease. The referral hospital was included in this study due to its specialized role in managing complex cases of DR-TB, including those with severe comorbidities and the need for advanced diagnostic tools. By selecting these facilities, this study aimed to represent the entire healthcare continuum in this high-burden region, allowing for a comprehensive assessment of treatment outcomes at different levels of care. The baseline data collected for this study included patient demographics (such as age and gender), clinical characteristics (including HIV status, BMI, and DR-TB type), treatment regimen details, and social history (like smoking status and comorbidities). This comprehensive data collection provided a solid foundation for analyzing the relationships between these factors and treatment outcomes.

Initially, a total of 456 patient files were reviewed. However, 96 files were excluded due to missing data on critical variables necessary for the analysis, including DR-TB treatment outcomes, HIV status, or BMI. The remaining 360 files had complete and consistent information across all required variables, enabling a thorough evaluation of this study objectives. Although this study included a total of 360 patient files with complete baseline data, not all files contained information for every outcome variable. Specifically, 212 files included laboratory results for culture conversion, 207 contained follow-up information on treatment duration, and 211 documented final treatment outcomes. The discrepancies in data availability arose from missing or incomplete records: some patients lacked culture conversion data due to incomplete testing or unavailable records; time in treatment data was missing for others who were transferred out or lost to follow-up; and treatment outcomes were not documented for cases where treatment was ongoing at the time of data collection or where reporting was incomplete. Out of the 360 patient files with complete baseline data, culture conversion, time in treatment, and treatment outcomes were analyzed in subsets depending on the availability of data for each variable. Specifically, culture conversion was evaluated in 212 files with laboratory results, time in treatment was analyzed in 207 files with documented follow-up data, and treatment outcomes were assessed in 211 files where outcomes were recorded. The varying sample sizes reflect the lack of complete data for some variables, such as unavailable laboratory results, patients who were transferred out, or ongoing treatment at the time of data collection.

### 2.2. Participant Selection Criteria

This study included all patient clinic files for individuals aged 18 and older who met the following criteria: they had complete information on drug-resistant tuberculosis (DR-TB) outcomes after starting a DR-TB treatment regimen, were HIV-positive and undergoing treatment, and had a body mass index (BMI) classification ranging from moderate to severe according to BMI indicators.

### 2.3. Data Analysis

Data were analyzed using IBM Statistical Package Social Sciences Statistics version 29 software. In this study, ChatGPT version 2, developed by OpenAI, was used for data analysis and visualization, and Python version 3.8. and R version 4.1.1 software was also used. Descriptive statistics were employed to summarize patient demographics, clinical characteristics, and BMI distributions. Comparative analyses were conducted to assess associations between BMI categories, HIV status, and DR-TB treatment outcomes. Statistical tests such as chi-square tests were used for categorical variables, and *t*-tests or ANOVA were applied for continuous variables where appropriate. Significance was determined at a *p*-value threshold of <0.05. Additionally, logistic regression analysis was performed to identify predictors of treatment outcomes, controlling for potential confounders.

## 3. Results

This study included 360 participants, including 156 women (43.3%) and 204 men (56.7%). Among the participants, the percentages of males (51.5%) and females (41.7%) classified as having a “Normal” BMI were calculated within their respective gender groups. These percentages reflect the proportion of males and females falling into the “Normal” BMI category compared to the total number of males (n = 204) and females (n = 156). Additionally, the data revealed that 14.2% of males and 8.3% of females were categorized as “Underweight”. These percentages were also calculated within each gender group. This finding contrasts with typical BMI distributions, which usually show a higher prevalence of underweight individuals among females. However, it highlights the unique characteristics of this study population. Factors such as the high prevalence of tuberculosis (TB) and its associated nutritional effects in this context may contribute to this gender disparity. This study identifies several factors that explain the differences in treatment outcomes for drug-resistant tuberculosis (DR-TB) between genders, with females showing higher cure rates than males. A key factor is health-seeking behavior; women tend to seek healthcare earlier and adhere more consistently to treatment regimens, facilitating early diagnosis and improving outcomes. Additionally, women may benefit from stronger social support systems that are essential for maintaining adherence to the lengthy and demanding DR-TB treatment process. In contrast, men often face greater challenges, such as comorbidities related to smoking, alcohol use, and diabetes. These conditions can adversely affect TB treatment efficacy by weakening immune responses or interfering with therapy. Moreover, men frequently encounter significant occupational and economic pressures, which can lead to interruptions in treatment adherence and poorer overall outcomes. These factors illustrate the complex interplay between behavioral, social, and health-related influences that contribute to the gender disparities observed in treatment success. This study also highlighted that multiple interrelated factors, including body mass index (BMI) and the type of drug-resistant tuberculosis (DR-TB), influenced the treatment outcomes of patients living with HIV. HIV-positive patients were more commonly associated with DR-TB types 1.0 and 2.0; however, their treatment outcomes were generally poorer compared to HIV-negative patients. This finding supports the existing evidence that HIV co-infection weakens immune responses, complicates treatment, and increases the likelihood of adverse outcomes, such as treatment failure, death, or loss of follow-up. Furthermore, the analysis emphasized the combined influence of HIV status, BMI, and DR-TB type on treatment success. For instance, HIV-positive individuals with a normal BMI had relatively better outcomes compared to those classified as thin or underweight, as malnutrition significantly exacerbated poor treatment results. Thin and underweight HIV-positive patients faced substantial challenges, highlighting the critical role of nutritional status in achieving treatment success. In contrast, HIV-negative patients demonstrated slightly better cure rates overall, particularly in DR-TB types where HIV status was less influential.

Most individuals tend to have a normal BMI range; however, during middle age (31–40), both overweight and underweight conditions become more common. This age group may be a critical target for specific nutritional interventions. Additionally, it is important to address underweight issues in older adults (51–60) to promote healthy aging (Figure 1).

Higher Proportion of “Normal” BMI Among Males: Within the male demographic (Gender = 1), the predominant proportion (51.5%) is classified within the “Normal” BMI category. Conversely, females (Gender = 2) also exhibit a notable representation (41.7%) in the “Normal” BMI category, albeit at a lower rate compared to their male counterparts. This observation implies that males exhibit a higher likelihood of achieving a “Normal” BMI as compared to females, which may suggest underlying lifestyle or metabolic disparities between the sexes. Greater Representation of “Obese” and “Overweight” Categories Among Females: The female cohort demonstrates an elevated percentage within the “Obese” (16.0%) and “Overweight” (20.5%) classifications in contrast to males, who exhibit 3.9% and 13.2%, respectively, in these categories. This indicates that females within this sample are more prone to attaining higher BMI classifications, potentially attributable to a multitude of factors encompassing biological, lifestyle, or hormonal distinctions that influence body composition. “Underweight” Category More Common Among Males: Within the “Underweight” classification, males constitute a larger segment (14.2%) relative to females (8.3%). This observation may suggest that males are marginally more susceptible to being categorized as underweight within this sample, potentially reflecting variances in dietary practices, metabolic rates, or occupational influences that affect body weight differentially across genders. “Thinness” Category Shows Slight Gender Difference: Both male and female populations exhibit comparable percentages in the “Thinness” category, with males at 13.2% and females at 13.5%. This nearly equivalent distribution indicates that the incidence of thinness is evenly apportioned across genders, suggesting that being marginally below the “Normal” BMI threshold does not significantly diverge by gender within this particular group (Figure 2).

Higher HIV-Positive Representation in DR-TB types 1.0 and 2.0: For DR-TB type 1.0, HIV-positive individuals make up 66.7% of cases, while HIV-negative individuals represent 33.3%. Similarly, in DR-TB type 2.0, 59.2% of cases are HIV-positive, and 40.8% are HIV-negative. These percentages indicate a strong association between HIV-positive status and DR-TB types 1.0 and 2.0. This could suggest that people living with HIV are more susceptible to these types of drug-resistant TB or that these types are more common among HIV-positive populations. Balanced Distribution in DR-TB types 3.0 and 4.0: In DR-TB type 3.0, the distribution is nearly even, with 52.4% HIV-positive and 47.6% HIV-negative individuals. DR-TB type 4.0 also shows a closer balance, with 62.5% HIV-positive and 37.5% HIV-negative. The relatively balanced distribution in these types implies that HIV status is less of a distinguishing factor for these DR-TB types, and they affect both HIV-positive and HIV-negative populations in similar proportions. High HIV-Negative Representation in DR-TB type 5.0: DR-TB type 5.0 has the highest percentage of HIV-negative individuals at 75.0%, with only 25.0% of cases being HIV-positive. This suggests a stronger association of DR-TB type 5.0 with the HIV-negative population, indicating that this type of drug-resistant TB may be less common among individuals living with HIV. Summary of Trends: Stronger Association of HIV-Positive Status with DR-TB types 1.0 and 2.0: The higher percentages of HIV-positive individuals in these DR-TB types suggest that these drug-resistant TB strains are more prevalent among HIV-positive populations. This may be due to immune system vulnerabilities associated with HIV, making individuals more susceptible to these specific types of drug-resistant TB. More Balanced HIV Status in Types 3.0 and 4.0: The closer balance of HIV-positive and HIV-negative individuals in DR-TB types 3.0 and 4.0 indicates that these types do not show a strong preference for either group, affecting both populations relatively equally. Predominantly HIV-Negative in DR-TB type 5.0: The high proportion of HIV-negative individuals in DR-TB type 5.0 could imply that this type is more likely to occur in individuals without HIV, suggesting different pathways or risk factors for this type of DR-TB as demonstrated by Figure 3.

Greater Cure Rate for Female Patients: The cure rate for female patients (Gender = 2) is 39.1%, greater than the cure rate for male patients (Gender = 1), which is 31.4% (Figure 4). This implies that, in comparison to male patients, female patients may react more favorably to DR-TB therapy or may have greater treatment adherence or results. Several variables, including biological differences, social support, and health-seeking behavior, may cause the variation in cure rates. Male Cure Rate: Although it is lower than that of female patients, the cure rate for male patients is still a respectable percentage of cases that are successfully treated, at 31.4%. Males had a lower cure rate, which would suggest that they face particular obstacles or difficulties in receiving effective therapy. Disparities in social variables that affect treatment adherence and results, greater incidence of comorbidities, or delayed healthcare-seeking behavior are some potential causes. Possible Factors Influencing Gender Differences in Cure Rates: Biological and Immunological Differences: Females may possess specific biological or immunological advantages that enhance their responses to treatment. Empirical studies have demonstrated that gender-based variations in immune response can significantly affect susceptibility to and recovery from infections, including tuberculosis (TB). Health-Seeking behavior: It has been observed that females generally exhibit a propensity to seek healthcare more swiftly than their male counterparts, which may facilitate earlier diagnosis and treatment, ultimately leading to improved cure rates. Conversely, males may exhibit delays in seeking medical care, potentially resulting in adverse outcomes. Social and Economic Factors: Variations in social support networks or economic stability across genders can significantly influence treatment adherence. Females may benefit from more robust support systems that promote compliance with prolonged TB treatments, whereas males may encounter obstacles related to occupational responsibilities, financial pressures, or societal stigma that hinder their ability to complete treatment. Lifestyle and Comorbidities: Lifestyle determinants, including tobacco use, alcohol consumption, and the prevalence of comorbid conditions (e.g., diabetes), are generally more pronounced in males, which may negatively correlate with cure rates. Such factors can compromise immune function or adversely interact with TB pharmacotherapy, thereby diminishing treatment efficacy. Summary: The data presented in the chart reveal a significant gender disparity in drug-resistant tuberculosis (DR-TB) cure rates, with females demonstrating a superior cure rate (39.1%) in comparison to males (31.4%). This discrepancy may stem from an interplay of biological, social, and behavioral determinants that influence treatment adherence and overall success. Acknowledging this gender disparity could enable healthcare providers to customize interventions more effectively, such as by offering enhanced support for male patients to elevate their cure rates.

Individuals classified as “Obese” and “Underweight” demonstrate the highest cure rates among all BMI categories, with rates of 42.4% and 42.9%, respectively (Figure 5). These results clearly establish that both high and low extremes of BMI are linked to a significantly increased likelihood of successful treatment. It indicates that individuals with substantial body reserves (the obese) or those who are more likely to adhere to medical treatments (the underweight) are better positioned to respond favorably to DR-TB therapy, although the precise reasons behind this relationship warrant further investigation. In contrast, individuals in the “Normal” BMI category achieve a cure rate of 34.7%, while those in the “Overweight” category display a slightly higher rate of 35.6%. These moderate figures reveal that those within the “Normal” or “Overweight” BMI ranges do not experience the same level of treatment success as those who fall at the extremes of the BMI spectrum. This asserts that factors beyond BMI play a more decisive role in influencing cure rates for these individuals. The “Thinness” category exhibits the lowest cure rate at 23.2%, significantly trailing the rates of other BMI groups. This strongly suggests that individuals with low body weight, indicative of malnutrition or other health complications, face substantial obstacles in achieving successful DR-TB treatment outcomes. Their low body weight critically impacts their immune response and tolerance to intensive TB medications, which directly contributes to their lower cure rates. In summary, the trends are clear: both “Obese” and “Underweight” individuals achieve the highest cure rates, demonstrating that BMI extremes positively influence treatment outcomes. More research is necessary to uncover the underlying mechanisms at play. Individuals categorized as “Normal” and “Overweight” exhibit only moderate cure rates, emphasizing that average body weight alone is not a significant factor in DR-TB treatment success. The “Thinness” group, with the lowest cure rate, underscores the increased vulnerability of low-BMI individuals in managing DR-TB. Underweight individuals must receive additional nutritional or supportive care to improve their outcomes. This analysis unequivocally illustrates that BMI is a crucial predictive factor for DR-TB treatment success. Patients at the BMI extremes (“Obese” and “Underweight”) benefit from higher cure rates, while those in the “Thinness” category face formidable challenges. Healthcare providers must consider BMI as an essential component of patient assessments when developing DR-TB treatment plans. For those identified as “Thin”, implementing further nutritional support or specialized care is imperative to enhance their treatment success.

Prevalence of “Normal” BMI Across Most DR-TB types: The “Normal” BMI category undeniably constitutes a significant portion of each type of DR-TB, particularly in DR-TB types 1.0 and 2.0. In these categories, individuals with a “Normal” BMI account for nearly half of the population, with 46.9% in DR-TB type 1.0 and 50.0% in DR-TB type 2.0. This clearly indicates that individuals with a “Normal” BMI are prominently represented among most DR-TB types, reflecting the widespread nature of DR-TB among those with average body weight ranges. Underweight Individuals Prominent in Specific DR-TB types: In DR-TB types 3.0 and 4.0, there is a marked prevalence of individuals categorized as “Underweight”, making up 31.2% and 25.0%, respectively. This strong correlation suggests a direct link between these DR-TB types and lower body weight, likely due to the severity of the disease or related nutritional challenges. Underweight individuals are particularly susceptible to these specific forms of drug-resistant TB. Obese and Overweight Categories in DR-TB types 1.0 and 2.0: DR-TB types 1.0 and 2.0 distinctly showed significant proportions of individuals classified as “Obese” and “Overweight”. For instance, in DR-TB type 1.0, 13.1% fall into the obese category, while 21.2% are classified as overweight. Similarly, DR-TB type 2.0 reveals 9.9% of individuals as obese and 12.5% as overweight. This evidence strongly suggests that higher BMI categories, such as obesity and overweight, are more prevalent in these DR-TB types, indicating that individuals with increased body weight experience unique interactions with these forms of drug-resistant TB. Thinness Category Shows Variability: The “Thinness” category clearly exhibited varied representation across different DR-TB types. DR-TB type 2.0 stands out with the highest rate of “Thinness” at 15.1%, while this category is less significant in others. This strongly indicates that thin individuals have a greater association with DR-TB type 2.0, though they are not widely represented in other DR-TB types, revealing the variability in the relationship between thinness and specific types of drug resistance. Balanced Distribution in DR-TB type 5.0: DR-TB type 5.0 presents an almost equal distribution across several BMI categories, with “Normal”, “Obese”, and “Underweight” each comprising 25.0%. This balanced distribution suggests that DR-TB type 5.0 does not correlate strongly with any specific BMI category, affecting individuals of diverse body weights more evenly. These data unequivocally reveal a connection between BMI and certain DR-TB types: Normal BMI stands out as the most common across all DR-TB types, indicating a baseline vulnerability to DR-TB, regardless of BMI. Underweight individuals are particularly prominent in DR-TB types 3.0 and 4.0, demonstrating that these types disproportionately impact those with lower body weight. Furthermore, those categorized as Obese and Overweight have a higher representation in DR-TB types 1.0 and 2.0, confirming the stronger association of these bodyweight categories with these specific DR-TB types. In contrast, DR-TB type 5.0 shows an even distribution of BMI, asserting a potentially equal impact across various BMI categories (Figure 6).

Dominance of “Normal” BMI: A significant portion of individuals diagnosed with Drug-Resistant Tuberculosis (DR-TB) fall within the “Normal” BMI range (Figure 7). For instance, in DR-TB types 1.0 and 2.0, individuals with a “Normal” BMI represent 46.9% and 50.0%, respectively. This prevalence suggested that “Normal” BMI may serve as an important baseline for understanding patient demographics across various DR-TB types. Variation in Obesity and Overweight: The presence of “Obese” and “Overweight” individuals was noteworthy, especially in DR-TB types 1.0, 2.0, and 3.0. In DR-TB type 2.0 alone, 9.9% of individuals were classified as obese, while 12.5% were categorized as overweight. These figures pointed to a substantial cohort affected by heightened BMI in these specific DR-TB types, highlighting the need for tailored treatment strategies that consider these patients’ unique health profiles. Underweight Prevalence in DR-TB types 3.0 and 4.0: There was a concerning trend of higher underweight proportions in DR-TB types 3.0 and 4.0, with figures of 31.2% and 25.0%, respectively. This observation suggested a potential link between these types of DR-TB and lower body weight, which could stem from the disease’s severity or related factors that contribute to weight loss. Understanding this relationship can help inform better nutritional support and management strategies for affected individuals. Thinness Across DR-TB Types: The “Thinness” category was particularly prominent in DR-TB type 2.0, with 15.1% of individuals affected. Conversely, other types displayed a lower prevalence of thin individuals. This pattern points to the possibility that thinness may be more closely associated with DR-TB type 2.0, emphasizing the need for focused research and intervention efforts in this area. Insights on DR-TB Type 5.0: In striking contrast, DR-TB type 5.0 demonstrates an even distribution across all BMI categories, with each category representing either 25.0% or no presence. This balanced distribution may suggest that DR-TB type 5.0 affects individuals across a diverse range of body weights, indicating the importance of broadening our understanding of how various factors influence susceptibility to different DR-TB types. Summary: The findings highlight that individuals with a “Normal” BMI are the most prevalent among all DR-TB types, serving as a key focus for further study. However, the higher percentages of underweight individuals in DR-TB types 3.0 and 4.0 merit attention, as they suggest a potential connection between these TB types and reduced body weight. Meanwhile, the prevalence of higher BMIs in types 1.0 and 2.0 reinforces the importance of tailored approaches to treatment. By carefully examining the distribution of BMI categories, health professionals can better devise strategies to address the unique needs of individuals affected by drug-resistant tuberculosis, ultimately improving patient outcomes and care.

The heatmap displays the Cramér V values representing the associations between DR-TB type, BMI, and HIV status (Figure 8). The analysis using Cramér V values revealed weak associations between DR-TB type, BMI, and HIV status. These weak associations suggest that BMI and HIV status may not significantly influence the distribution of DR-TB types in this study population. However, these factors were selected for analysis based on their well-documented biological and clinical relevance in TB management. BMI is a critical indicator of nutritional status, which can impact immune response and treatment adherence, while HIV status is a known driver of TB susceptibility and progression. The weak associations observed might be influenced by unmeasured confounders, such as socioeconomic factors, treatment adherence, and the severity of comorbidities, which were not included in the analysis. Additionally, the heterogeneity within this study population, including varying stages of disease progression and differences in access to care, may have diluted potential associations. These limitations underscore the need for further research to explore these relationships in greater depth, incorporating a broader range of variables. Despite the weak associations, the analysis highlights the importance of considering these factors as a part of a comprehensive approach to DR-TB management. The findings reinforce the need for interventions that address nutritional support and HIV care, while also prompting further investigation into the interplay of these and other factors in shaping DR-TB outcomes.

### Definition of Operational Terms

A heatmap is a two-dimensional visualization of data that uses color to represent numerical values. This can be either different intensities of the same hue or different colors from a palette [23].

Cramér’s V is a crucial metric in statistics and data analysis that evaluates the degree of correlation between two categorical variables. This coefficient, which is derived from the chi-square statistic, has a normalized value in the range of 0 to 1, where 1 denotes an ideal link and 0 denotes no association [23].

The following definitions were defined according to Merriam-Webster Medical Dictionary [24]:

Nutritional status refers to evaluating an individual’s health in terms of their diet, body weight, and biochemical data. It is commonly assessed using clinical indicators such as body weight, BMI, and the rate of unintentional weight loss.

Overweight and obesity are defined as abnormal or excessive fat accumulation that presents a health risk. A body mass index (BMI) over 25 is considered overweight, and over 30 is obese.

Synergy in medicine is when an interaction of two or more drugs is described, their combined effect is greater than the sum of the effects seen when each drug is given alone.

Thinness is generally defined as a physical condition in which the body mass index (BMI) is below 18.5.

Hypertension (high blood pressure) is when the pressure in your blood vessels is too high (140/90 mmHg or higher). It is common but can be serious if not treated.

Diabetes mellitus is impaired insulin secretion and variable degrees of peripheral insulin resistance leading to hyperglycemia.

Cancer is a large group of diseases with one thing in common: they happen when normal cells become cancerous cells that multiply and spread.

Kidney disease means your kidneys are not working properly and are beginning to lose their function.

Liver disease refers to liver disease, and they are usually referring to chronic conditions that cause progressive damage to your liver over time. Viral infections, toxic poisoning, and certain metabolic conditions are among the common causes of chronic liver disease.

Mental illness, also called mental health disorders, refers to a wide range of mental health conditions—disorders that affect your mood, thinking, and behavior.

Co-infection is the simultaneous infection of a host by multiple pathogen species. In virology, co-infection includes the simultaneous infection of a single cell by two or more virus particles [25].

Comorbidities are medical conditions that coexist alongside a primary diagnosis and affect your health, including your treatment and outlook. Common comorbidities among hospitalized people include TB and HIV [25].

The National TB and Leprosy Control Program standard [1,25], World Health Organization (WHO) definitions, and the 2021 modification of the WHO definitions and reporting framework for TB treatment outcomes used the following definitions:

Cured: a TB patient who started treatment with bacteriologically verified TB and finished the treatment as advised by a national policy with evidence of a bacteriological response but no signs of failure.

Treatment completed: a TB patient who followed the national policy suggested a course of therapy but whose results did not fulfill the criteria for cure or treatment failure.

Treatment failure: a TB patient whose treatment plan required being stopped or permanently switched to another treatment plan.

Died: a TB patient who passed away for whatever reason, either before beginning treatment or while receiving it.

Lost to follow-up: a TB patient who has not started therapy or whose regimen has been stopped for eight or more weeks in a row after starting treatment at least four weeks previously.

Not evaluated: a TB patient for whom no treatment result has been established. This applies to situations where the reporting unit is unsure of the treatment outcome and situations where the case has been transferred to another treatment facility.

New patients: Patients with TB who have never undergone TB therapy or who have just started receiving anti-TB drugs. New patients may have positive or negative bacteriology and may have disease in any area of the body.

Previously treated: a TB patient may have the disease at any anatomical site, positive or negative bacteriology, and have taken anti-TB medications for at least one month in the past.

The following two treatment outcomes are classified by World Health Organization standards [2]:

Successful treatment outcome: if TB patients finished therapy with symptom clearance or were cured (i.e., had a negative smear microscopy at the end of treatment and on at least one prior follow-up test).

Unsuccessful treatment outcome: if TB patients had treatment but had treatment failure (i.e., were still smear-positive after five months), lost to follow-up (i.e., patients who stopped taking their medication for two or more months consecutively after registering), or died.

## 4. Discussion

We conducted a retrospective review of clinical data from the files of TB-infected patients aged 18 and older who had completed treatment and were co-infected with HIV. The baseline data were meticulously collected from patient files at four TB clinics and a referral hospital associated with these clinics in the rural areas of the Olivier Reginald Tambo District Municipality in the Eastern Cape, South Africa. We carefully analyzed patient demographics, clinical characteristics, and social history from the collected data. Additionally, we thoroughly examined culture conversion rates, treatment durations, and treatment outcomes based on the data from 360 patient files. These patients had a body mass index (BMI) that indicated a moderate-to-severe classification. BMI is a factor influencing treatment outcomes for DR-TB because it is a reliable indicator of a patient’s nutritional status, which is closely linked to TB treatment success [21]. Malnutrition, reflected by low BMI, weakens the immune system and impairs the body’s ability to respond effectively to tuberculosis treatment [6,7]. Conversely, obesity (high BMI) can complicate treatment due to associated metabolic conditions, inflammation, or the need for medication adjustments [10]. Prior studies have demonstrated that both underweight and obesity can impact outcomes differently, with underweight individuals facing higher mortality rates and obese individuals encountering unique challenges [16]. In resource-limited settings, such as the rural Eastern Cape, malnutrition and BMI extremes (underweight or obese) are prevalent challenges that exacerbate DR-TB treatment [2]. Monitoring BMI allows identifying at-risk patients who may benefit from additional nutritional or supportive interventions to enhance treatment success. Furthermore, WHO recognizes nutritional support as a critical component of TB care [2]. BMI is a simple, cost-effective tool to assess and address nutritional needs alongside DR-TB treatment. BMI is a modifiable factor, unlike fixed variables such as age or gender. BMI-related challenges can be addressed through targeted interventions like dietary support and nutritional supplementation, directly improving treatment efficacy. Thus, BMI was chosen in our study for its well-established role as a predictor of treatment success and its ability to highlight nutritional vulnerabilities that significantly influence DR-TB outcomes. This study revealed that, while most individuals had a normal BMI, middle-aged patients (31–40 years old) were more likely to exhibit both overweight and underweight conditions. This age group could be important for targeted nutritional interventions [26,27]. Additionally, thinness in older adults (51–60) may require further attention to ensure healthy aging [28,29]. This study demonstrated a clear association between DR-TB types and HIV status, with HIV-positive individuals being more affected by DR-TB, especially the more common RR and MDR types [30]. The DR-TB patients successfully demonstrated higher percentages of treatment outcomes in patients with comorbidities such as hypertension and type 2 diabetes mellitus, and the majority of these patients were cured or completed their treatment, and these findings are consistent with a study conducted in India [31]. DR-TB patients with mental illness and liver disease showed less favorable outcomes, with lower percentages of successful treatments. Furthermore, they were associated with higher rates of adverse outcomes, death, and loss of follow-up [32]. DR-TB patients with comorbidities such as cancer and kidney disease (KD) also demonstrated significant challenges, with higher rates of non-successful treatment outcomes [33]. Therefore, the overall trends noted in this study demonstrated that the presence of comorbidities with conditions generally leads to a more complex treatment landscape for DR-TB, with some conditions like mental illness and liver disease significantly affecting the likelihood of a successful outcome. This variability in outcomes was based on different comorbidity conditions suggesting that these health issues should be carefully managed alongside DR-TB treatment. This study’s analysis suggested that gender plays a role in DR-TB treatment outcomes, with females showing a marginally better cure rate compared to males. This difference could be influenced by a variety of factors, including biological, social, or treatment-related variables [34]. The evaluation of drug-resistant tuberculosis treatment performed in this study clearly showed that patients with a normal BMI have the best cure rates. At the same time, those who are underweight or suffering from thinness are at a disadvantage in terms of treatment success [35,36]. Therefore, this highlights the importance of maintaining adequate nutritional status during DR-TB treatment, especially for patients with lower BMI, as malnutrition could adversely affect their recovery. This study noted a mix of successful and adverse outcomes, with a significant portion of HIV-positive males achieving a cure or completing treatment. However, the rates of loss to follow-up and death are notable, indicating areas where additional support and intervention might be needed as indicated. These findings are similarly shared by other studies elsewhere [37,38]. This study highlighted that normal BMI generally correlates with better treatment outcomes across all age groups. However, older patients showed a gradual increase in adverse outcomes. Further, the Underweight and Thinness categories in this study consistently showed poorer outcomes across all age groups, with the risk increasing with age [39,40]. The Overweight and Obese individuals show varied outcomes depending on age, with older individuals facing more challenges that are significant. This analysis highlighted the compounded effect of age and BMI on DR-TB treatment outcomes, particularly emphasizing the vulnerability of older, underweight individuals, and this study’s findings are consistent with the findings of other studies [41,42]. Nutritional interventions and age-specific strategies may be crucial in improving treatment success. There was a synergy of DR-TB type, BMI category, and HIV status [35]. This study showed the distribution of different BMI categories across the DR-TB types (RR, MDR, Pre-XDR) and highlighted how these categories are split between HIV-positive and HIV-negative individuals. This phenomenon has been demonstrated by studies elsewhere [35,43,44]. The key insight in this study is that normal BMI dominates across all DR-TB types, particularly in HIV-positive individuals with RR-TB [45,46]. Thinness and Underweight were more prevalent among HIV-positive individuals, especially in the RR-TB type. In addition, the Overweight and Obese categories have a smaller representation but are present across different DR-TB types and HIV statuses. In our study, HIV-positive individuals tended to have a higher representation in most DR-TB types, especially within the normal and thin BMI categories [47]. HIV-negative individuals are fewer in number but are still present across all DR-TB types and BMI categories [46]. The interaction between these factors is clear, showing that the distribution of individuals varies significantly based on both BMI and HIV status across different types of DR-TB. The heatmap visualizes the Cramér’s V values for the association between each pair of variables (DR-TB type, BMI, and HIV status), and all three pairs demonstrated a weak association, suggesting that these factors do not have a strong interrelationship [28].

### 4.1. Limitations

It is important to acknowledge that our study has some limitations. We only had access to clinical data from TB-infected patients collected from the files at four TB clinics and one referral hospital associated with these clinics in rural areas of the O.R. Tambo District Municipality. Consequently, not all clinics in the municipality were included in our study. Furthermore, significant variables, such as comorbidities, distance to health facilities, income level, and behavioral factors (including knowledge and attitudes about TB), as well as smoking and alcohol history, were not consistently recorded in the TB registers. Additionally, our study concentrated solely on TB outcomes and did not consider the outcomes of other diseases. However, the strength of this study lies in its demonstration of the challenges associated with treatment outcomes in TB management, particularly for patients co-infected with HIV and those with differing nutritional statuses, such as underweight versus overweight.

### 4.2. Recommendations and Further Studies

This study emphasizes the importance of conducting similar research in various settings across the Eastern Cape, involving all age groups while ensuring adequate funding and time allocation. The current study focused on the Oliver Reginald Tambo Region, which faces significant challenges related to tuberculosis (TB), HIV, and nutritional status. Additionally, it is recommended that TB treatment strategies include nutritional interventions along with routine HIV screening and management, considering the synergistic relationship between these diseases.

## 5. Conclusions

Our findings indicate that sociodemographic factors, including age and gender, significantly influence treatment outcomes for DR-TB patients. Middle-aged individuals (31–40 years) were notably more prone to dual challenges of underweight and overweight conditions, suggesting a critical need for targeted nutritional interventions in this group. Additionally, older adults (51–60 years) with underweight status require focused attention to promote healthy aging. Gender disparities were evident, with females showing higher cure rates than males, potentially due to differences in health-seeking behaviors and social support systems. Multiple factors, including comorbid conditions, BMI categories, and HIV status, influence treatment outcomes in DR-TB. Patients with extreme BMI values, such as underweight or obese, demonstrated higher cure rates, underscoring the role of nutritional status in enhancing immune response and treatment adherence. Conversely, the thinness category exhibited the lowest cure rates, emphasizing this subgroup’s need for nutritional support. Comorbidities like mental health challenges, liver disease, and cancer further complicated treatment outcomes, while patients with comorbidities such as hypertension or diabetes showed comparatively better success rates. Gender plays a pivotal role in DR-TB treatment outcomes, as evidenced by the higher cure rates observed among female patients compared to their male counterparts. This difference can be attributed to biological and immunological advantages in females, more proactive health-seeking behaviors, and stronger social support systems. Male patients, on the other hand, often face barriers such as delayed healthcare access, higher rates of comorbidities, and lifestyle factors like smoking or alcohol use, which may hinder treatment adherence and success. These findings highlight the necessity for gender-specific interventions to address these disparities.

## Figures and Tables

**Figure 1 ijerph-22-00319-f001:**
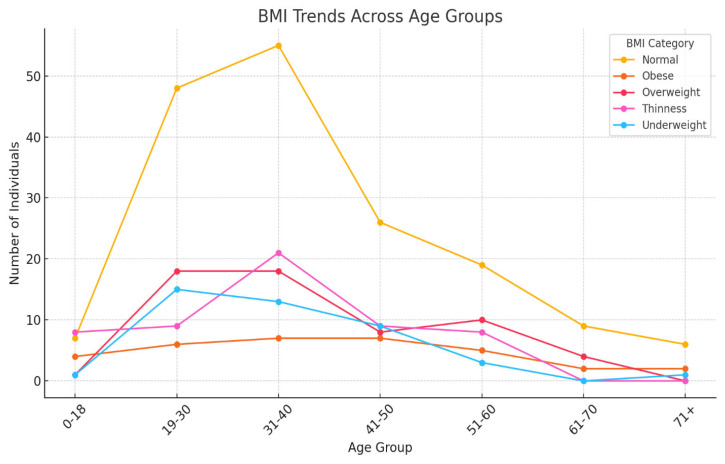
The line chart shows BMI trends across age groups.

**Figure 2 ijerph-22-00319-f002:**
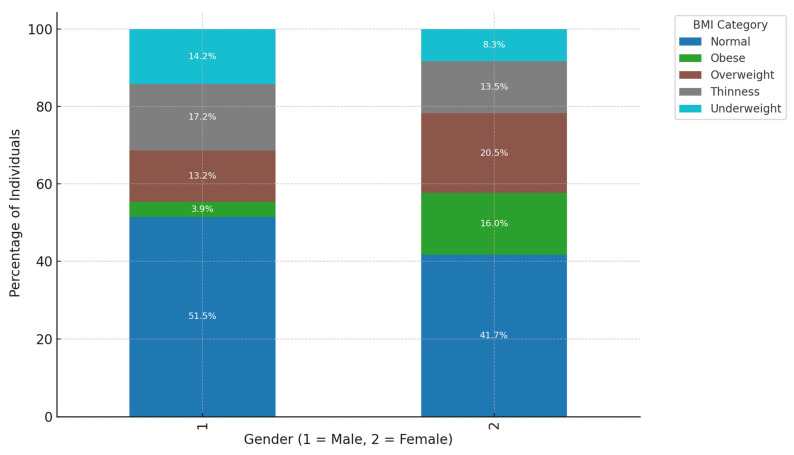
The bar chart illustrates the distribution of BMI categories across genders.

**Figure 3 ijerph-22-00319-f003:**
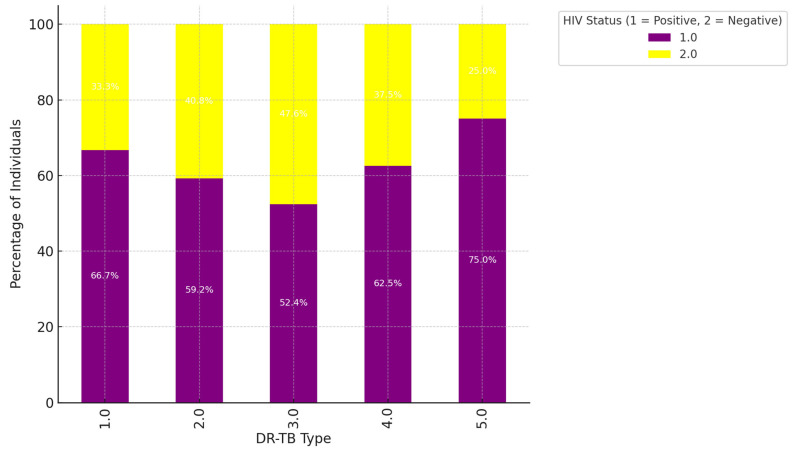
The bar chart visualizes the distribution of different DR-TB types by HIV status.

**Figure 4 ijerph-22-00319-f004:**
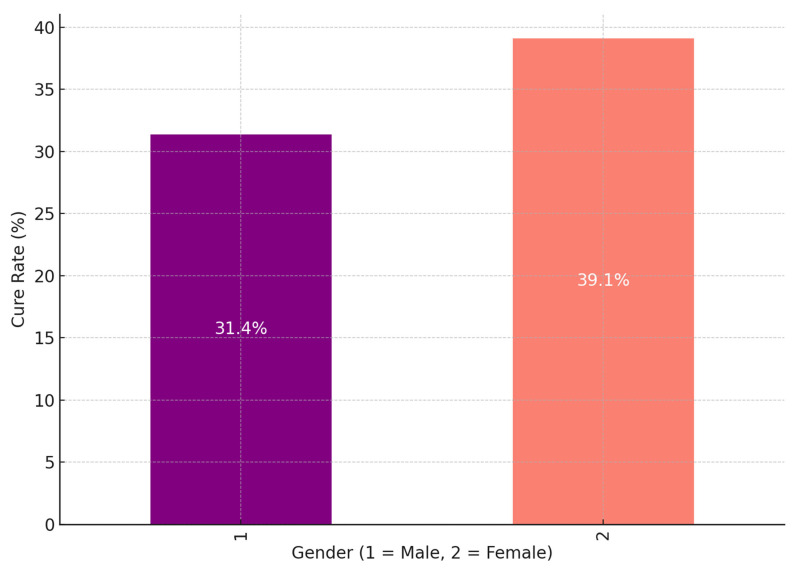
The bar chart shows the DR-TB cure rates by gender.

**Figure 5 ijerph-22-00319-f005:**
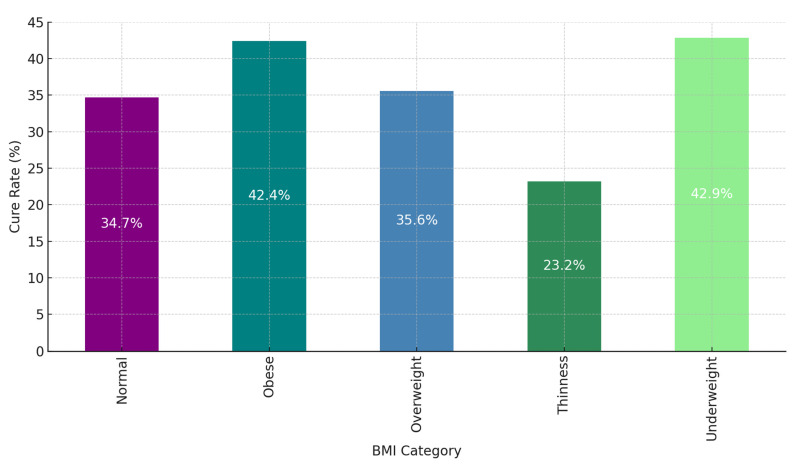
The bar chart illustrates the impact of BMI on cure rates for patients with drug-resistant tuberculosis (DR-TB).

**Figure 6 ijerph-22-00319-f006:**
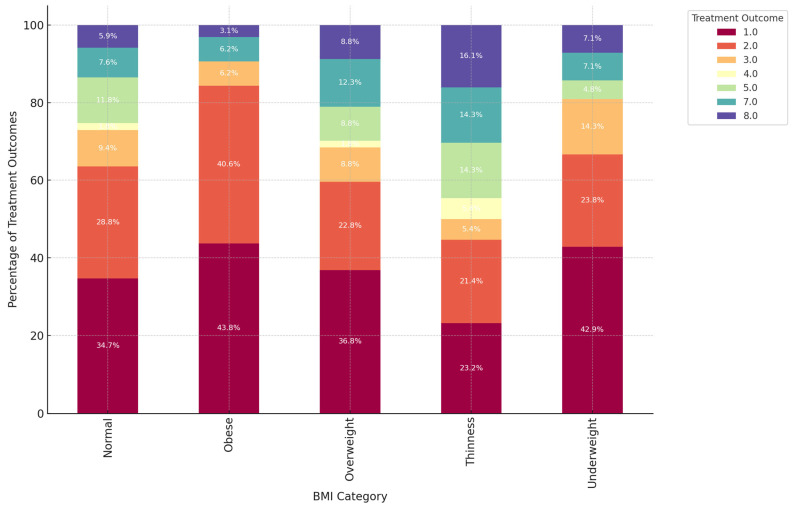
The bar chart illustrates the relationship between BMI, age groups, and treatment outcomes for DR-TB patients.

**Figure 7 ijerph-22-00319-f007:**
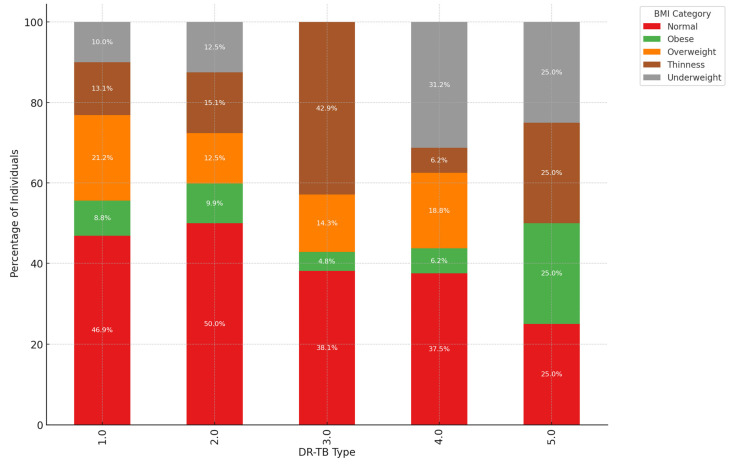
The synergy between DR-TB type, BMI category, and HIV status.

**Figure 8 ijerph-22-00319-f008:**
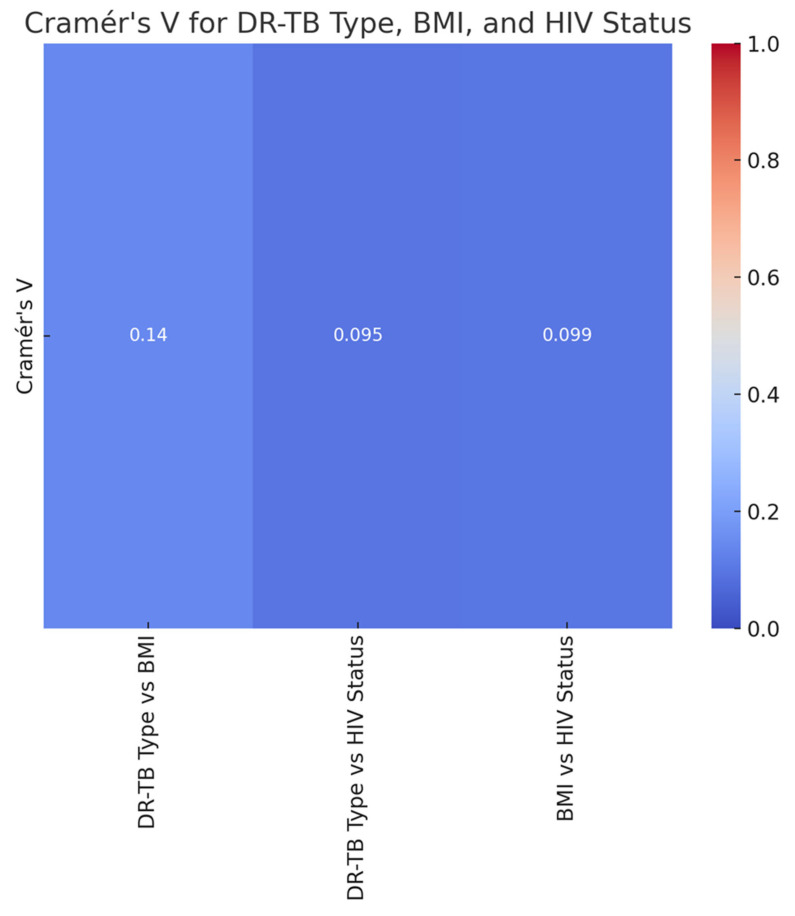
The heatmap visualization of the Cramér’s V values for the association between each pair of variables (DR-TB type, BMI, and HIV status).

## Data Availability

The datasets generated and analyzed during the current study are available from the corresponding author upon reasonable request.

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
