# Peer review of "Outcomes of Treating Tuberculosis Patients with Drug-Resistant Tuberculosis, Human Immunodeficiency Virus, and Nutritional Status: The Combined Impact of Triple Challenges in Rural Eastern Cape"

_ijerph, 2025, doi:10.3390/ijerph22030319_

Round 1
Reviewer 1 Report
Comments and Suggestions for Authors
OUTCOMES OF TREATING TUBERCULOSIS PATIENTS WITH DRUG-RESISTANT TUBERCULOSIS, HUMAN IM MUNODEFICIENCY VIRUS, AND NUTRITIONAL STATUS: THE COMBINED IMPACT OF TRIPLE CHALLENGES IN RURAL EASTERN CAPE
Dear author and editor:
The article talked about different patterns drug resistant TB patients and treatments, and the impact of HIV and nutrition on the treatment outcomes. The article could be published after a minor revision. I have some comments on it:
· The legends of all figures have to be written better with more details.
· In the figure 4. Why the cure rates of female patients higher than male patients?
· Why the author choose the MBI as factor influence treatment outcomes?
· It is not clear in the conclusion if the HIV patients have better treatment outcomes.
Thank you very much, best regards
Author Response
Dear Editor,
I am attaching the reviewer's comments for reviewer 1.
Thank you.
Dr. Dlatu

Reviewer 2 Report
Comments and Suggestions for Authors
Abstract
1. The abstract uses inconsistent terminology in the results section for both "drug-resistant tuberculosis" (DR-TB) and "Type 1 (RR-TB)" / "Type 2 (MDR-TB)". While it is common to refer to different forms of DR-TB, the distinction between "Type 1" and "Type 2" is not universally used in this context. It would be clearer to consistently use the standard terminology: RR-TB (rifampicin-resistant tuberculosis) and MDR-TB (multi-drug-resistant tuberculosis), or more precisely explain what "Type 1" and "Type 2" refer to.
2. The results section describes "overweight and underweight individuals among middle-aged and elderly people," but it seems to be more focused on BMI distribution rather than treatment outcomes. The abstract should ensure that the results are more closely aligned with the treatment outcomes and co-morbidities (HIV, nutritional status, etc.) that were central to the study. The connection between BMI and treatment outcomes should be clarified.
3. The distribution of BMI across different age groups and genders seems out of place in the results section unless it is clearly linked to TB treatment outcomes. The focus should be on how BMI impacts TB treatment success or failure, or other related outcomes.
4. The sentence "it suggests limited connections between DR-TB type, BMI category, and HIV status" is vague. The abstract does not specify what "limited connections" mean. It would be helpful to indicate whether there was no significant correlation or if any trends were observed, even if they were weak.
5. The abstract mentions using a scatter plot to display BMI data. Scatter plots are typically used for showing relationships between two continuous variables, but BMI is a categorical variable (e.g., underweight, normal weight, overweight, obese). It would be more appropriate to use bar charts or box plots for categorical data. If the scatter plot is indeed used, a brief explanation of how it was used to assess trends would be helpful.
6. The conclusion focuses on "prioritizing the management of co-morbid conditions in DR-TB patients," but this is not well-supported by the results. The abstract does not present strong evidence of how co-morbidities (like HIV or malnutrition) directly impacted treatment outcomes. The conclusion should be more tightly linked to the results of the study.
Introduction
1. There is a lot of repetitive phrases in the introduction. Authors revisit certain points multiple times, leading to redundancy. For instance: The relationship between TB and HIV is mentioned multiple times, especially in the sentences discussing how each exacerbates the other. The role of obesity in exacerbating TB and HIV is repeated across several sentences (e.g., obesity exacerbates the impact of TB and HIV and obesity complicates the course of these diseases). These points could be condensed and reorganized for better flow and clarity. I strongly suggest that authors should streamline these concepts into fewer, more concise statements to avoid overemphasis and to make the narrative more focused.
2. From line 70 – 87, while authors adequately provided the definitions of TB resistance types (MDR-TB, XDR-TB, etc.), the flow is disrupted by the detailed explanation of these terms. The explanation of drug-resistant TB (DR-TB) and its various categories is somewhat technical and may lose general readers. My suggestion is summarize the main categories briefly and consider putting the detailed explanation of resistance in a footnote or a parenthetical statement for readability. For example: DR-TB, including MDR-TB and XDR-TB, arises when Mycobacterium tuberculosis becomes resistant to first-line and second-line TB drugs, complicating treatment efforts [12].
3. The flow of the introduction is somewhat disjointed from line 39 – 47, page 1 & 2. The shift from the WHO's positive statement on TB treatment to the challenges of TB, HIV, and obesity in South Africa seems abrupt. There is a jump between sections focusing on the individual diseases to their interplay and treatment challenges, making it a bit hard to follow the narrative from one topic to the next.
4. In page 2, line 70 73, the term "triple burden" is introduced as referring to the co-occurrence of DR-TB, HIV, and obesity. However, the introduction does not explain why "triple burden" is the most appropriate term or how it connects to the broader context. The term might be confusing unless further context is given about its use globally or within public health discourse.
5. The introduction describes the general challenges posed by TB, HIV, and obesity, but it could do more to tie these issues specifically to the study's setting in the Eastern Cape. Alos, the mention of "rural areas of Eastern Cape" appears late in the introduction, and more context could be provided earlier on to frame why this region is significant for the study.
6. The introduction places a strong emphasis on socioeconomic factors and malnutrition in the early paragraphs. While these factors are important, the opening section could be more concise and focused. The mention of "poverty being an important determinant of both problems" appears twice within a few sentences.
7. The introduction is quite long and dense. While it covers a broad range of relevant topics, the length could make it overwhelming for readers. A more structured breakdown with shorter paragraphs would enhance readability.
8. The introduction does a good job of outlining the problem and the context of the study, but the research gap—the reason for this particular study—could be highlighted more clearly. Although the conclusion mentions the lack of research in rural Eastern Cape, this could be emphasized earlier as the driving force behind the study.
9. The introduction mentions complex relationships between TB, HIV, and obesity, but the causal or synergistic connections are not fully explained. For example, how does obesity exactly worsen the treatment outcomes of TB and HIV? Is it due to immune suppression, metabolic effects, or treatment complications?
10. There are a lot of citations in this section, and some references are listed in ways that might make the reading a bit clunky (e.g., "[5]" and "[10]" in the middle of sentences).
Methods
1. In page 3, line 132-134, the phrase "retrospectively evaluated clinical data" is generally clear, but it could be more specific about the time frame of the data collection. Was it a review of patient files from a specific year or range of years?
- The mention of "four TB clinics and one referral hospital linked to the clinics" is good, but more details about the setting would improve clarity. How are these clinics selected, and why is one referral hospital included? Including these details can better contextualize the research.
- Specify the time period of the data evaluation and briefly explain the selection of the clinics and hospital for better context.
- The selection criteria mention a BMI range from "moderate to severe" but does not specify what these categories refer to in terms of BMI cutoffs. What exactly is the classification for "moderate to severe" such as BMI classified as moderate or severe (BMI ≤ 18.5 for underweight, 18.5–24.9 for normal weight, 25–29.9 for overweight, and ≥30 for obesity)? how did you determine that? This needs clarification.
- The criteria also include patients who are both HIV-positive and on treatment. However, it would be helpful to specify whether the patients had confirmed co-infection with DR-TB and HIV, or if these were considered separately.
- The phrase "baseline data was collected" is vague. It would be better to specify what aspects of baseline data were collected, such as age, gender, clinical history, treatment regimen, etc.
- The mention of "456 patient files reviewed, 360 contained complete information" raises the question of why 96 files were excluded. Were they missing data on critical variables like DR-TB outcomes, HIV status, or BMI? Explaining why certain files were excluded would make the methodology clearer.
- The numbers for "culture conversion (n=212)", "time in treatment (n=207)", and "treatment outcomes (n=211)" seem inconsistent. Should the total number of patients in each of these categories be the same, if all the missing values are removed? There’s a discrepancy between the total number of complete files (360) and the breakdown for each of these factors. This discrepancy needs clarification.
- It is not clear whether culture conversion, time in treatment, and treatment outcomes are measured in a subset of the 360 patient files or across all of them. Specifying this would avoid confusion.
- The formula for BMI is correctly mentioned, but it may be useful to state whether BMI was recorded directly from the patient files or calculated using the height and weight data. If the latter, was the BMI classification derived from guidelines (e.g., WHO guidelines) or another standard? For example: BMI was calculated using the formula BMI = weight (kg) / height² (m²) based on recorded patient weight and height, and classified according to the World Health Organization (WHO) standards.
- The section does not mention the statistical analysis methods used. How were the results analysed (e.g., using software like SPSS, R, or other tools)? Were any statistical tests applied to assess the significance of findings?
- The section uses both "DR-TB" and "multidrug-resistant tuberculosis (DRTB)" interchangeably, which could be confusing. Consistency in terminology is important.
Results
1. The statement "the average age for the 17 individuals is 32" is unclear. The study involves 360 participants, but the average age is stated for only 17 individuals. If this is meant to represent a subgroup, it should be clearly specified. The average age of all 360 participants should be mentioned if that is the focus, or clarify why only 17 individuals are being referenced.
2. Authors failed to use appropriate statistical measures (e.g., average, standard deviation) for the entire cohort and provide more context around the subgroup analysis. The biography characteristics of the participants should be summarised in table format for clarity.
3. The total gender distribution (43.3% women and 56.7% men) adds up to 100%, which is correct. However, it's unclear whether the number of men and women corresponds to these percentages or if other data could clarify this. If the total is 360 participants, then there should be 156 women and 204 men, but it’s worth confirming if these numbers match the actual dataset.
4. The "Higher Proportion of 'Normal' BMI Among Males" section claims that 51.5% of males are classified as "Normal," and 41.7% of females fall into the same category. However, these values are described as percentages within the gender subgroups but don’t specify how these percentages are calculated (i.e., are these percentages relative to all participants, or are they within the respective gender subgroups?).
5. "Underweight" Category More Common Among Males also claims that 14.2% of males are underweight compared to 8.3% of females. This seems inconsistent with the distribution presented elsewhere (inconsistent with the typical BMI distribution where underweight would more commonly be seen in females), so it’s worth reviewing the data to ensure this is accurate.
6. The analysis using Cramér V values suggests weak associations between DR-TB type, BMI, and HIV status. While this statistical insight is useful, more detailed interpretation of these weak associations is needed. Weak associations may indicate that BMI and HIV status don’t significantly influence DR-TB type, but further clarification is needed as to why these factors were selected for the analysis. Are other confounders not accounted for, and how does this affect the interpretation of the results?
7. The summary of trends section appears to repeat earlier findings about HIV status and BMI but lacks detailed statistical analysis (such as p-values or effect sizes) to support these conclusions. These claims about the relationships between HIV status and DR-TB types need to be backed up with statistical tests to ensure they are not simply descriptive.
8. While the results provide percentages and comparisons (e.g., male vs. female cure rates, BMI categories, HIV status), there is no mention of any statistical tests (such as t-tests, chi-square tests, or ANOVAs) to assess whether the observed differences are statistically significant. Without these tests, it is impossible to conclude whether the differences between groups are likely due to chance or represent real effects. It would be important to perform and report results from appropriate statistical tests to validate whether the observed differences in cure rates, BMI distributions, and HIV status across DR-TB types are significant.
Author Response
Dear Editor,
Please see the attached reviewer comments for reviewer 2.
Thank you.
Dr. Dlatu

Reviewer 3 Report
Comments and Suggestions for Authors
Dear authors,
I would like to thank you for the opportunity to review this manuscript and say that this is an important topic for public health, especially in patients with drug-resistant tuberculosis, infected with the human immunodeficiency virus and with impaired nutritional status.
Abstract: The abstract did not clarify the study period. Better emphasize the main results of the study and what can be done to improve the outcomes of TB patients.
Introduction:
The study provides a good theoretical framework relating nutritional status to patients with HIV and TB. Infections exacerbate nutritional deficiencies, which in turn increase morbidity and mortality from infectious diseases. Adequate nutrition plays a critically important role in supporting the health and quality of life of people with TB and human immunodeficiency virus.
Methods:
I suggest providing more details about the study location in order to justify why these places were chosen.
Results
The results are well described and the operational terms are well defined.
Discussion
I suggest removing the first sentence and being more objective in the discussion, as this has already been described in the methodology: "We conducted a retrospective review of clinical data from the files of TB-infected patients aged 18 and older who had completed treatment and were co-infected with HIV". The discussion text contains a lot of description of methods. Emphasize the main findings and discuss based on the literature. I suggest discussing with data extracted from the literature a paragraph with each topic of the conclusion:
-sociodemographic profile
-factors influencing treatment outcome
-discuss why gender plays an important role in treatment outcomes
Author Response
Dear Editor,
Please see the attached reviewer's comments for reviewer 3.
Thank you.
Dr. Dlatu

Round 2
Reviewer 2 Report
Comments and Suggestions for Authors
1. This revised version of the manuscript is slightly more concise, maintains all the critical information, and improves the clarity and flow.
2. This version of introduction is more concise, improving clarity and eliminating redundancies, while retaining the important context and justifications for the study. It should be more accessible for readers and set up the study's importance in a clearer, more structured way.
3. The method has been improved to have a clear and logical flow between sections enhances the readability, and providing more specifics about handling missing data and statistical methods adds transparency to the analysis.